# Leveraging RGB-Pressure for Whole-body Human-to-Humanoid Motion Imitation

## ABSTRACT

Whole-body motion imitation has gained wide attention in recent years as it can enhance the locomotive capabilities of humanoid robots. In this task, non-intrusive human motion capturing with RGB cameras is commonly used for its low-cost, efficiency, portability and user-friendliness. However, RGB based methods always faces the problem of depth ambiguity, leading to inaccurate and unstable imitation. Accordingly, we propose to introduce pressure sensor into the non-intrusive humanoid motion imitation system for two considerations: first, pressure can be used to estimate the contact relationship and interaction force between human and the ground, which play a key role in the balancing and stabilizing motion; second, pressure can be measured in the manner of almost non-intrusive approach, which can keep the experience of human demonstrator. In this paper, we establish a RGB-Pressure (RGB-P) based humanoid imitation system, achieving accurate and stable end-to-end mapping from human body models to robot control parameters. Specifically, we use RGB camera to capture human posture and pressure insoles to measure the underfoot pressure during the movements of human demonstrator. Then, a constraint relationship between pressure and pose is studied to refine the estimated pose according to the support modes and balance mechanism, thereby enhancing consistency between human and robot motions. Experimental results demonstrate that fusing RGB and pressure can enhance overall robot motion execution performance by improving stability while maintaining imitation similarity.

## KEYWORDS

Motion Imitation, Humanoid Robot, Multi-modal Fusion, Motion Retargeting

## 1 INTRODUCTION

Humanoid have long been a focal point of robotics research, embodying engineering challenges related to human biology, cognition, and motor abilities [54]. Despite their human-like appearance often suggests higher interactivity and approachability compared to other forms of robots, traditional control methods relying predefined action sequences limits the adaptability and autonomy of robots in real-world environments [27, 47]. To address this limitation and broaden the range of actions achievable by humanoid robots, researchers have introduced motion imitation as a means to

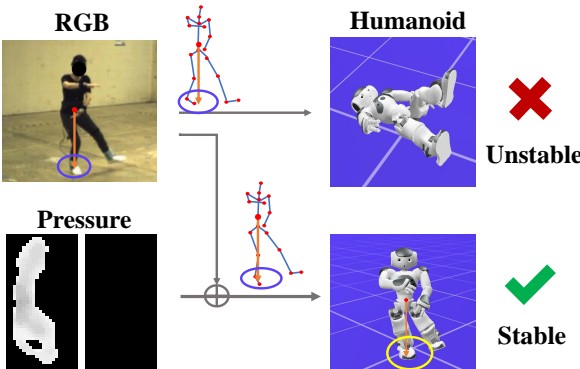

**Figure 1: Motivation of our method.** When performing an action where the CoM projection on the ground exceeds the foot support region, the robot falls. However, when the action, corrected by the pressure data, is input to the robot, the action is executed stably.

enhance their capabilities, facilitating humanoid interaction with the physical world in a manner similar to humans [11].

Nonetheless, there is the issue of creating a precise, efficient, and user-friendly demonstration format for human demonstrators during imitation [29, 34]. Previous methods have employed manipulators [20, 21], force feedback devices (exoskeleton) [2, 19], and high-precision motion capture equipment such as inertial motion capture [12, 24, 43] and optical motion capture systems [10, 18]. However, these tools are often expensive, operationally complex, poor portability, and cumbersome to wear. In contrast, low-cost, non-intrusive devices like RGB or RGB-D cameras offer significant advantages [5, 16, 45, 59, 67].

Regrettably, Methods relying solely on RGB often encounter challenges in accurately capturing 3D representation of human movements due to depth ambiguity and uncertainties in the foot-ground contact relationship, resulting in motion that is unstable and potentially hazardous for robots. Fig. 1 depicts a classic movement within Tai Chi, known as the single-leg stance. The pose estimated from the RGB image perceptually describes the tilted state of the human body, but inaccurately determines the center of mass (CoM), leading to an unreasonable reference. When introducing pressure for guidance, the reference pose can be adjusted to incorporate a more reasonable CoM. This correction will greatly enhance the ability of humanoid robots to imitate human motion effectively.

In this paper, we propose a novel non-intrusive method for whole-body human-to-humanoid motion imitation by integrating RGB and pressure information. Initially, we establish a systematic baseline comprising three modules: pose estimation, motion retargeting, and whole-body control. This system not only captures human poses in the real world but also retargets them into the new pose space of

a humanoid robot, utilizing them to drive the robot in reality. Additionally, foot pressure data is obtained through pressure sensors embedded in insoles. Subsequently, the estimated pose undergoes refinement by determining the support mode and correcting the center of mass (CoM). To validate the efficacy of our proposed method, we compare the motion imitation performance of RGB and RGB-P based methods using similarity and stability metrics. Furthermore, we showcase motion imitation results not only in simulation but also in real-world scenarios. Through our endeavors, we demonstrate that integrating RGB and pressure information can facilitate human-to-humanoid motion imitation efficiently, accurately, and safely while maintaining user-friendliness.

The contributions of our work can be summarized as follows:

- We explore the feasibility and superiority of the pressure modality for non-intrusive robot motion imitation through both theoretical analysis and experimental validation.
- We develop a non-intrusive human-to-humanoid motion imitation system with RGB and pressure.
- We evaluate the performance not only in simulation but also in real environments, demonstrating that the proposed method enhances the completeness and stability of robot motion imitation tasks while ensuring motion similarity.

## 2 RELATED WORK

### 2.1 Physics-Based and Multi-Modal Human Motion Capture

The rapid development of monocular motion capture has attracted more and more widespread attention [6, 23, 26, 30, 41, 55]. However, due to the depth ambiguity of monocular images, the estimated human motion does not meet the real physical constraints. In particular, the principle of balance is not satisfied.

To enhance the realism of virtual human movements, some researchers have incorporated physical constraints and corrections [31, 44, 49, 50, 60, 61]. These endeavors provide valuable priors for estimating accurate human body poses.

On the basis of monocular motion capture, adding multi-modal information has also become a method to improve the plausibility and stability of virtual human. Von et al. [56] uses offline optimization method to estimate human body pose by fusing RGB and IMUs. While, Liang et al. [32] and Pan et al. [40] combine IMUs and human body 2D keypoints to obtain pose and translation. Recently researchers have paid attention to the importance of pressure for motion capture and tried to introduce pressure as supervision or constraint information in monocular motion capture. Scott et al. [46] curated a dataset containing real human action images, poses, and foot pressure data. However, their emphasis was primarily on estimating pressure from human body images. Tripathi et al. [55] achieves more precise motion capture by inferring physical characteristics of the human body, such as center of mass (CoM), center of pressure (CoP) and contact pressure, and constructs a human body dataset containing pressure for evaluation. Zhang et al. [63] introduced pressure as supervision to enhance the estimation of virtual human contacts, thereby obtaining more accurate human pose and translation. Pressure has emerged as our preferred non-invasive tool, offering a wealth of physical information.

## 2.2 Humanoid Teleoperation by Imitating

The teleoperation of humanoids through imitation has been prevalent for a considerable period [3, 11]. He et al. [16] categorized it into three types: task space teleoperation [2, 8, 9, 48, 65], upper-body teleoperation [4, 14, 58, 62, 66], and whole-body teleoperation [5, 12, 13, 16–18, 24, 38, 52, 53, 67]. We focus on the third type to explore the scalability of humanoid robots in whole-body motion capabilities.

When transferring human motion to humanoid robots, the selection of motion capture modalities and motion representation is crucial for bridging the gap between humans and robots. The challenge is to identify the most relevant features that capture the essence of human actions. Various approaches have been explored in this area, including using CoM [2, 8, 13, 38], joint rotations or positions [4, 12, 14, 67], force [19, 37], and other information for representation and mapping. He et al. [16] utilized image-based motion capture for teleoperation of humanoids, representing a commendable effort towards user-friendly teleoperation. However, their tracking of the lower body of the robot was not sufficiently precise. These uni-modal methods didn't integrate information from multiple dimensions. This leads to the loss of human motion features, and sometimes can impose redundant information unsuitable for imitation onto the robot. Whole-body teleoperation based on multi-modal motion capture is relatively rare. It is worth mentioning that Dafarra et al. [9] integrated IMUs, pressure insoles, and optical sensors into their iCub3 avatar system, achieving effective teleoperation in task space. However, the complex wearing devices are not user-friendly for operators, and they do not emphasize the consistency and similarity of lower-body motion between humanoid robots and humans.

In whole-body motion imitation, a crucial issue is how to balance the similarity and stability of the lower body. There are two approaches for mapping lower-body motions: (1) Robot-centric [12, 18, 42, 59]: This approach emphasizes robot control, where the human posture is transferred to the robot. Here, the robot autonomously selects or predefines the appropriate support based on the actual posture situation. (2) Human-centric [24, 28, 67]: In contrast, this approach prioritizes human support, where the support is obtained from the human and then mapped to the robot. Subsequently, the robot adjusts its posture based on the acquired support mode. These two approaches give dominance to one side without unifying the consistency of human-robot actions. Either humans make stiff leg movements to accommodate the robot's structure, or robots experience delays to adapt to human actions, significantly increasing the risk of instability, especially when the posture conflicts with the support mode.

Therefore, we believe that Image-based motion capture can provide a good user experience, while integrating pressure sensing can ensure consistency between support and posture, thereby unifying human and humanoid motion.

## 3 OVERVIEW

The ultimate goal of human-to-humanoid motion imitation is that humanoid robots can perform exactly the same motion as humans, i.e., $\tilde{\mathbb{P}}_{Robot} \doteq \tilde{\mathbb{P}}_{Human}$. However, achieving end-to-end human-to-humanoid motion imitation is not a straightforward task and

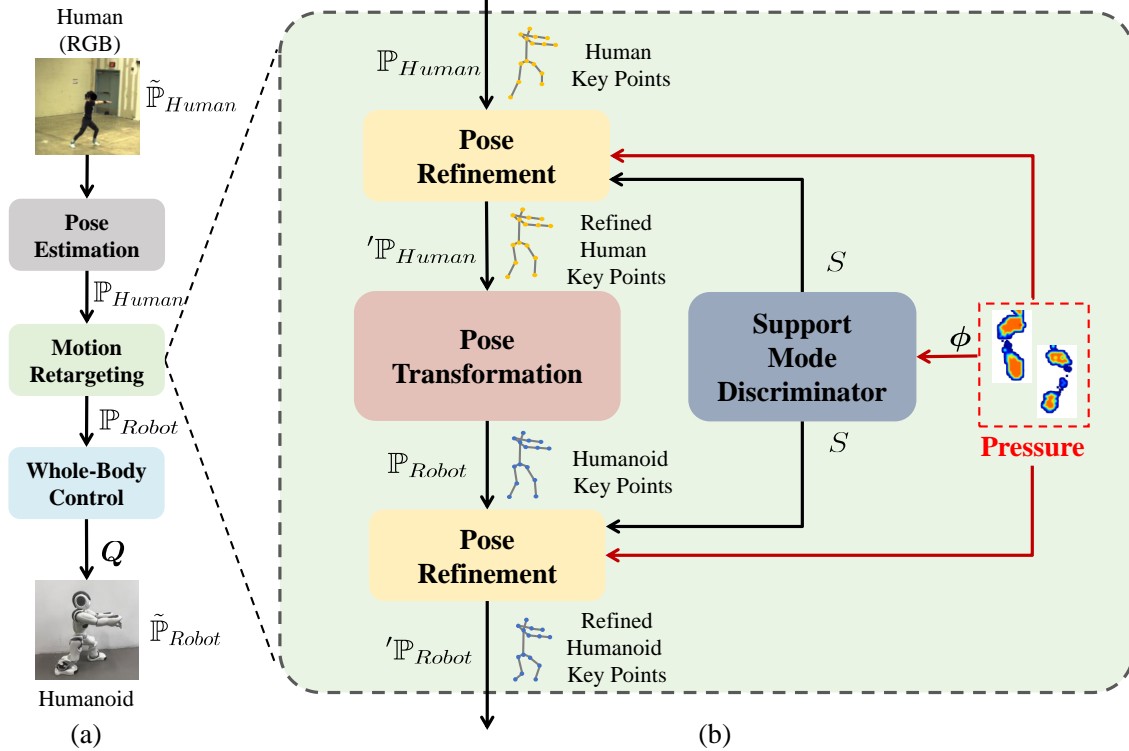

**Figure 2: Overview of our method.** (a) The framework of human-to-humanoid motion imitation. (b) Our proposed motion retargeting leveraging pressure.

typically involves three main steps, as illustrated in Fig. 2 (a). **Pose Estimation** serves as the initial step in capturing and representing the human pose in the real world $\tilde{\mathbb{P}}_{Human}$, into an interpretable and executable format $\mathbb{P}_{Human}$. Since there are still differences in size, degree of freedom, and structure between human and humanoid robot, **Motion Retargeting** is necessary to transfer the estimated human body pose $\mathbb{P}_{Human}$ into humanoid robot pose $\mathbb{P}_{Robot}$. Considering the control strategy and balance constraints of the robot, it is crucial to determine precise drive parameters $Q$ within the **Whole-Body Control** module to represent the final pose in the real world $\tilde{\mathbb{P}}_{Robot}$. In this paper, we develop a human-to-humanoid motion imitation system with non-intrusive sensors (e.g., RGB cameras and pressure insoles). The details are elaborated as follows.

### 3.1 Pose Estimation

Pose estimation utilizing monocular RGB camera have emerged as the predominant approach, owing to their non-intrusive nature, cost-effectiveness, and convenience. However, due to the depth ambiguity of RGB images, estimating 3D joint points from RGB images is ill-posed. To solve this problem, we introduce the parametric human model SMPL [33] to offer a robust human structure prior in natural human poses. We obtain the SMPL model parameters $\Theta$ of the human body by leveraging the off-the-shelf monocular human pose estimation method CLIFF [30]. In this case, the sub-problem of pose estimation can be formulated as:

$$\min_{\Theta} \quad L_e(\mathbb{P}_{Human}(\Theta), \tilde{\mathbb{P}}_{Human}) \tag{1}$$

where the estimated human pose $\mathbb{P}_{Human}(\Theta)$ can be represented by $\{J_H(\Theta), M_H(\Theta)\}$. $J_H(\Theta) \in \mathbb{R}^{24 \times 3}$ represents the positions of 24 joints, including head, hands, elbows, and feet [33]. $M_H(\Theta) \in \mathbb{R}^{1 \times 3}$ is the position of human body's CoM [55]. Both of them are calculated by the SMPL model parameters $\Theta$. It is expected that $\mathbb{P}_{Human}(\Theta)$ is as similar as possible to $\tilde{\mathbb{P}}_{Human}$, which is measured by Euclidean distance $L_e(\cdot)$.

### 3.2 Motion Retargeting

As shown in Fig. 3, there are large differences between human and humanoid robot in terms of topology, quantity, size and structure, etc. Motion retargeting is a essential process in Human-to-Humanoid motion imitation. The estimated human pose $\mathbb{P}_{Human}(\Theta)$ undergoes a series of transformations, including reduction, scaling, and coordinate system alignment [24, 39], to yield the humanoid robot pose $\mathbb{P}_{Robot} = \{J_R, R_R, M_R\}$. Here, $J_R \in \mathbb{R}^{16 \times 3}$ denotes the positions of the 16 robot joints, $R_R \in \mathbb{R}^{16 \times 3 \times 3}$ denotes the rotation of the feet, while $M_R \in \mathbb{R}^{1 \times 3}$ represents the position of the humanoid robot's CoM.

### 3.3 Whole-Body Control

While the retargeted pose $\mathbb{P}_{Robot}$ exhibits a high degree of morphological similarity to the human pose, it cannot be directly applied

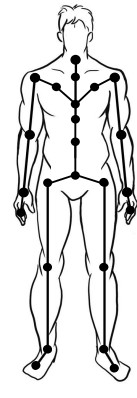 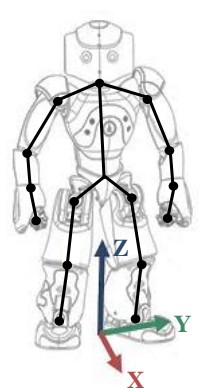

(a) Human with 24 joints     (b) Humanoid robot with 16 joints

**Figure 3: Differences between human and humanoid robot.**

to the humanoid entity due to the inability to meet real physical constraints, thus leading to potential issues such as falling or imbalance. To solve this problem, whole-body control is employed following the classical approaches [36]. Specifically, $\mathbb{P}_{Robot}$ is input into a differential inverse kinematics (IK) solver to calculate the joint control parameters $Q$ by utilizing the robot's Jacobian matrix under the constraints of robot's kinematic balance.

This can be formulated as

$$\min_{Q} \quad L_e(\mathbb{P}_{Robot}, \tilde{\mathbb{P}}_{Robot}(Q))$$
$$\text{s.t.} \quad \tilde{\mathbb{P}}_{Robot}(Q) \subseteq C_{robot}, \tag{2}$$

where $C_{robot}$ represents a set of stable and safe motion.

## 4 MOTION RETARGETING USING PRESSURE

As shown in Fig. 2 (a), the pose estimated from RGB encounters challenges related to depth ambiguity and an unclear foot-ground contact relationship. Little attention is paid to verifying whether the RGB-estimated human body key points align with reality, which significantly impacts the stability of robot imitation. To enhance the physical realism of human key points, both the CoM, representing the overall kinematic state, and the foot-ground contact, reflecting human balance control, play significant roles. Since representing them solely from RGB information is highly hard, we introduce a new modality, pressure, and represent the problem of robot motion imitation in the following form:

$$\min_{\Theta, Q} \quad L_e(\mathbb{P}_{Human}(\Theta), \tilde{\mathbb{P}}_{Human}) + L_e(\mathbb{P}_{Robot}, \tilde{\mathbb{P}}_{Robot}(Q))$$
$$\text{s.t.} \quad \mathbb{P}_{Robot} = \delta(\mathbb{P}_{Human}(\Theta), \phi) \tag{3}$$
$$\tilde{\mathbb{P}}_{Robot}(Q) \subseteq C_{robot}.$$

Combining Eq. 1 and 2, we introduce an additional constraint, where $\phi$ denotes the pressure between the human feet and the ground. It serves as a link within the function $\delta(\cdot)$ to establish the mapping between the human body model and the humanoid. Building upon the analysis, we elaborate on our motion retargeting method using pressure. As shown in Fig. 2 (b), the human key points is expanded

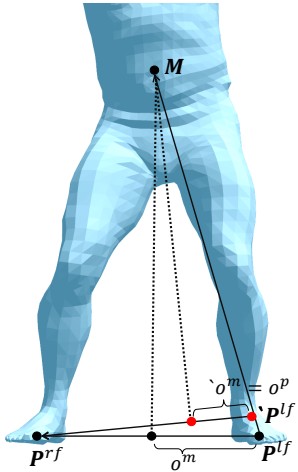

**Figure 4: CoM offset definition and pose refinement.**

to $\mathbb{P}_{Human}(\Theta) = \{J_H(\Theta), M_H(\Theta), P_H(\Theta)\}$, where $P_H(\Theta)$ is the position of CoP calculated by SMPL parameters $\Theta$ [55]. Additionally, we can acquire highly accurate foot-ground contact information based on pressure data, which is then fed into a Support Mode Discriminator to determine the current support mode $S$ of the human body. Then, $S$ and $\phi$ are used to refine the estimated pose $\mathbb{P}_{Human}$ and $\mathbb{P}_{Robot}$ respectively.

### 4.1 Stability Analysis

To incorporate pressure into human pose representation, let's begin by analyzing the stability of human and humanoid. Both humans and humanoids have two support areas (feet), each bearing a portion of the body's mass. The pressure insoles measure the pressure distribution on the soles of each foot, which is presented in pixels. From the pressure and positions of the pixels, we can obtain CoP of each foot $P = \{P^{lf}, P^{rf}\}$. As shown in Fig. 4, $(\cdot)^{lf}$ and $(\cdot)^{rf}$ means left and right foot. Taking the left foot as an example:

$$\phi^{lf} = \sum_{i \in lf} \phi_i, \tag{4}$$

$$P^{lf} = \frac{\sum_{i \in lf} X_i \cdot \phi_i}{\phi^{lf}}. \tag{5}$$

Here, $\phi_i$ represents the pressure value of each pixel, and $X_i$ is its position. $P^{lf}$ is the CoP of the left foot. The calculation method for the right foot is similar. In theory, we can also obtain the CoP of the whole body, written as

$$P = (1 - o^p) \cdot P^{lf} + o^p \cdot P^{rf}, \tag{6}$$

where $o^p$ is a pressure offset denoted as follows:

$$o^p = \frac{\phi_{rf}}{\phi_{lf} + \phi_{rf}}. \tag{7}$$

This value ranges from 0 to 1, with a value of 0 when all the pressure falls on the left foot and a value of 1 when all the pressure falls on the right foot.

Next, we follow [24, 38, 42] to extend the concept of CoM offset and define it as

$$o^m = \begin{cases} \frac{(M-P^{lf})(P^{rf}-P^{lf})}{\|P^{rf}-P^{lf}\|_2^2} & , \text{if } S = D \\ 0 & , \text{if } S = L \\ 1 & , \text{if } S = R. \end{cases} \quad (8)$$

Here, $S$ represents the support mode, $L$, $R$, and $D$ respectively denote left leg support, right leg support, and dual legs support. $o^m \in [0, 1]$. When the body is fully supported by the left leg, this value is 0, and when it is fully supported by the right leg, the value is 1.

Under quasi-static conditions, the CoP can be considered equivalent to the projection of the CoM onto the ground [7, 22, 57], so we have the following relationship when $S = D$:

$$\begin{aligned} o^m &= \frac{(P - P^{lf})(P^{rf} - P^{lf})}{\|P^{rf} - P^{lf}\|_2^2} \\ &= \frac{((1 - o^p)P^{lf} + o^p \cdot P^{rf} - P^{lf})(P^{rf} - P^{lf})}{\|P^{rf} - P^{lf}\|_2^2} \\ &= \frac{o^p(P^{rf} - P^{lf})(P^{rf} - P^{lf})}{\|P^{rf} - P^{lf}\|_2^2} \\ &= o^p. \end{aligned} \quad (9)$$

Eq. 9 indicates that under quasi-static conditions, the pressure offset $o^p$ is equal to the CoM offset $o^m$. Hence, we can utilize the pressure offset $o^p$ to correct the CoM offset $o^m$, ensuring that the human key points align with the pressure distribution.

## 4.2 Kinematic Pose Refinement

For a human demonstrator, we can derive an estimated $o^m$ from RGB image and a real $o^p$ from pressure data. However, the estimated $o^m$ and the $o^p$ are usually not equal. Following the analysis presented in Sec. 4.1, we apply a geometric method, as described in [24], to refine estimated pose, ensuring consistency between $o^m$ and $o^p$. Specifically, the pose of human $\mathbb{P}_{Human}$ and humanoid $\mathbb{P}_{Robot}$ are refined respectively according to the support mode (i.e., Dual support and single support).

**Dual support**. As illustrated in Fig. 4, the ultimate target of pose refinement is to find a new offset $'o^m$ satisfies $'o^m = o^p$. Assuming the right foot remains fixed, our goal is to locate a new left foot CoP $'P^{lf}$, which is constrained along the line connecting $M$ and $P^{lf}$. According to Eq. 8, we have:

$$'o^m = \frac{(M -' P^{lf})(P^{rf} -' P^{lf})}{\|P^{rf} -' P^{lf}\|_2^2}. \quad (10)$$

After solving the Eq. 10, we can obtain the refined CoP $'P^{lf}$. Then, the normal vector of the foot plane can be obtained as follow

$$n^{lf} = M - ('P^{lf} +' o^m(P^{rf} -' P^{lf})). \quad (11)$$

From the normal vector, we can obtain the orientation of the foot plane by

$$R^{lf} = \cos(\theta)I + (1 - \cos(\theta))n^{lf} \cdot (n^{lf})^T + \sin(\theta)[n^{lf}]_\times \quad (12)$$

where $I$ is the identity matrix, $[n_{lf}]_\times$ represents the skew-symmetric matrix of the normal vector, $n_{lf}$ is the angle between foot normal vector and ground normal vector.

---

**Algorithm 1:** Support Mode Discriminator Pseudo-code

**Input** : $S$, $o^p$
**Output**: $S$

1  **if** $S == L$ **then**
2    **if** $o^p > th$ **then**
3      $S \leftarrow D$
4  **else if** $S == R$ **then**
5    **if** $o^p < 1 - th$ **then**
6      $S \leftarrow D$
7  **else if** $S == D$ **then**
8    **if** $o^p \leq th$ **then**
9      $S \leftarrow L$
10   **else if** $o^p \geq 1 - th$ **then**
11     $S \leftarrow R$

---

The refinement process for the right foot follows the same principle. When $o^m > o^p$, we perform left foot refinement; otherwise, we perform right foot refinement.

**Single support**. When the support mode $S = L$ or $R$, there is no need to refine the position. It only needs to calculate the orientation of feet by current $P^{lf}$ or $P^{rf}$ according to Eq. 11 and 12.

Note that, orientation information is not crucial for refining the human body, as we do not focus on human joint orientation. However, it is vital for refining the robot key points, as the robot's foot plane must align with the normal vector to ensure balance.

Subsequently, the refined pose $'\mathbb{P}_{Robot}$ is fed into whole-body control module to drive the humanoid robot, facilitating the achievement of balanced and stable motion.

## 4.3 Support Mode Discriminator

There is little research that can precisely capture foot-ground contact using RGB, RGB-D, or even IMU data. However, pressure sensing presents notable advantages in this context, as it accurately captures changes in the body's CoP, thereby determining the human support mode. Our designed discriminator is illustrated in Algorithm. 1, where $th$ is the threshold for the pressure distribution. We employ the concept outlined in Eq. 7. When $o^p$ surpasses $th$, a switch in the support mode is activated.

We believe that in the process of action imitation, accurate mapping of leg support modes is crucial, as it constitutes the essence of imitation. Otherwise, in cases where only the overall location is considered, the leg structure of the humanoid robot would be meaningless.

## 5 EXPERIMENTS

To assess the efficacy of our proposed human-to-humanoid motion imitation method, we utilize the PSU Taiji MultiModal (PSU-TMM100) Dataset [46] as the human motion demonstrator and the NAO humanoid robot [15, 25, 51] as the motion executor. We quantitatively and subjectively compare the similarity and stability of methods based on RGB and RGB-P modalities.

| | All sequences | | | Normalized sequences | | |
|---|---|---|---|---|---|---|
| | $\mathfrak{E}_{mpjpe,H}\downarrow$ | $\mathfrak{E}_{mpjpe,R}\downarrow$ | $\mathfrak{E}_{frechet}\downarrow$ | $\mathfrak{E}_{mpjpe,H}\downarrow$ | $\mathfrak{E}_{mpjpe,R}\downarrow$ | $\mathfrak{E}_{frechet}\downarrow$ |
| RGB | 94.95 | 15.19 | 525.71 | 97.37 | 15.45 | 535.99 |
| RGB-P(Ours) | 95.76 | 16.01 | 532.53 | 98.76 | 15.93 | 539.43 |

Table 1: Quantitative results of similarity.

| | $\mathfrak{E}_{complete}\uparrow$ | All sequences | | Normalized sequences | |
|---|---|---|---|---|---|
| | | $\mathfrak{E}_{com}\downarrow$ | $\mathfrak{E}_{cop}\downarrow$ | $\mathfrak{E}_{com}\downarrow$ | $\mathfrak{E}_{cop}\downarrow$ |
| V-MoCap | 113001 | 35.37 | 44.86 | 34.87 | 47.32 |
| RGB | 82746 | 41.33 | 61.65 | 36.98 | 56.27 |
| RGB-P(Ours) | 96434 | 30.41 | 31.81 | 32.28 | 36.31 |

Table 2: Quantitative results of completeness and stability.

## 5.1 Experimental Setup

**Dataset.** The PSU Taiji MultiModal (PSU-TMM100) Dataset [46] comprises 100 Taiji motion sequences performed by 10 human subjects, providing RGB video and foot pressure. Additionally, a wearable optical motion capture system (Vicon motion capture system (V-MoCap) [1]) is used to obtain precise and accurate 3D markers on human body. We select this dataset for the following reasons: (1) It contains RGB and pressure modal data, aligning with the requirements of our method; (2) It offers accurate human body 3D marker data, serving as a good benchmark for non-intrusive pose estimation; (3) Tai Chi encompasses numerous balancing motions, posing significant challenges for humanoid robot.

**Platform.** NAO humanoid robot [15, 25, 51] has 25 degrees of freedom (DOF). Its motion model is based on generalized inverse kinematics and performs well in tasks involving Cartesian and joint control, balance, and other functions. Our simulation environment utilizes Webots and qiBullet. We evaluate the the similarity of humanoid in qiBullet and stability in Webots.

**Metrics.** We employ the Mean Per Joint Position Error (MPJPE) $\mathfrak{E}_{mpjpe}$ and the Frechet Distance $\mathfrak{E}_{frechet}$ for similarity evaluation, while for stability evaluation, we utilize Imitating Duration $\mathfrak{E}_{length}$, CoM Deviation $\mathfrak{E}_{com}$, and CoP Deviation $\mathfrak{E}_{cop}$.

1) MPJPE $\mathfrak{E}_{mpjpe}$ : After aligning the estimated and ground-truth 3D joints at the root, we calculate the MPJPE $\mathfrak{E}_{mpjpe}$ (mm) to measure the accuracy of the estimated pose.

2) Frechet Distance $\mathfrak{E}_{frechet}$: Due to the mismatch in the joint numbers and link sizes between human and humanoid, we use the root-aligned mean per-joint Frechet distance $\mathfrak{E}_{frechet}$ (mm) to evaluate the similarity of all motion joints of the robot and the corresponding human groundtruth [62], including the head, elbows, hands, and feet.

3) Imitating Duration $\mathfrak{E}_{complete}$: We evaluate the completeness of humanoid imitation by summing the total lengths of stable action sequences ($\mathfrak{E}_{complete}$) executed by the robot without experiencing falls. Throughout the experiment, we terminate the process whenever the absolute height of the head drops below 250 millimeters, indicating a fall by the robot.

4) CoM Deviation $\mathfrak{E}_{com}$: To evaluate the stability of the humanoid during execution, we measure the mean global deviation $\mathfrak{E}_{com}$

(mm) between the CoM projection and the ideal support region. Specifically, when the humanoid stands on dual legs, we calculate the distance from the CoM projection to the line between the ankle joints [39]. When the robot stands on a single leg, the distance is from the CoM projection to the ankle joint projection of the supporting leg.

5) CoP Deviation $\mathfrak{E}_{cop}$: We also use the whole-body CoP computed from the foot sensors. To calculate the global deviation of the CoP and ideal support region $\mathfrak{E}_{cop}$ (mm).

## 5.2 Similarity Evaluation

Considering that V-MoCap can provide precision and accurate 3D markers on human body, we follow [35] to obtain the ground-truth of human body key points $\tilde{\mathbb{P}}_{Human}$ through SMPL model [33]. So that, we can compare the motion imitation similarity of our proposed RGB-P based method with the RGB based method through $\mathfrak{E}_{mpjpe}$ and $\mathfrak{E}_{frechet}$. The results are demonstrated in Tab. 1. Given that robots cannot execute all human actions, the motion imitation always terminate early with falling down. For a fair and comprehensive comparison, we provide test results for two cases. The term of "All sequences" in the table indicates that both our method and the comparative method use their respective metrics at their highest completion. The term of "Normalized sequences" indicates normalizing the sequence lengths to a uniform length. For instance, if our RGB-P method achieves a completion of 2000 frames and the RGB method achieves a completion of 1000 frames, the metrics are calculated across all frames (i.e., 2000 for RGB-P method and 1000 for RGB method) in terms of "All sequences". Meanwhile, the metrics are calculated across the minimum number of frames (i.e., 1000 for both RGB-P and RGB methods) in terms of "Normalized sequences".

The $\mathfrak{E}_{mpjpe,H}$ in the first column primarily presents the error between the estimated human pose (i.e., $\mathbb{P}_{Human}$ for RGB method and $'\mathbb{P}_{Human}$ for RGB-P method) and the ground-truth $\tilde{\mathbb{P}}_{Human}$. $\mathfrak{E}_{mpjpe,R}$ represents the error between the target pose of the robot (i.e., $\mathbb{P}_{Robot}$ for RGB method and $'\mathbb{P}_{Robot}$ for RGB-P method) and the actual executed pose $\tilde{\mathbb{P}}_{Robot}$. For RGB method, there is no human key points refinement module, and support mode discriminator relies on a method from the estimated pose [28, 64].

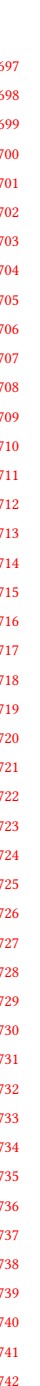

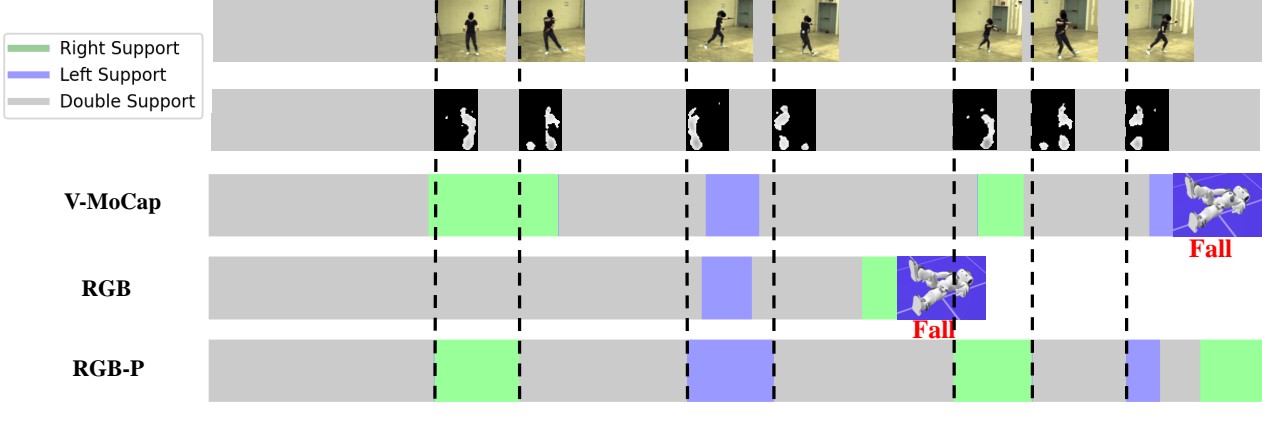

**Figure 5: Comparison of human support mode discrimination.**

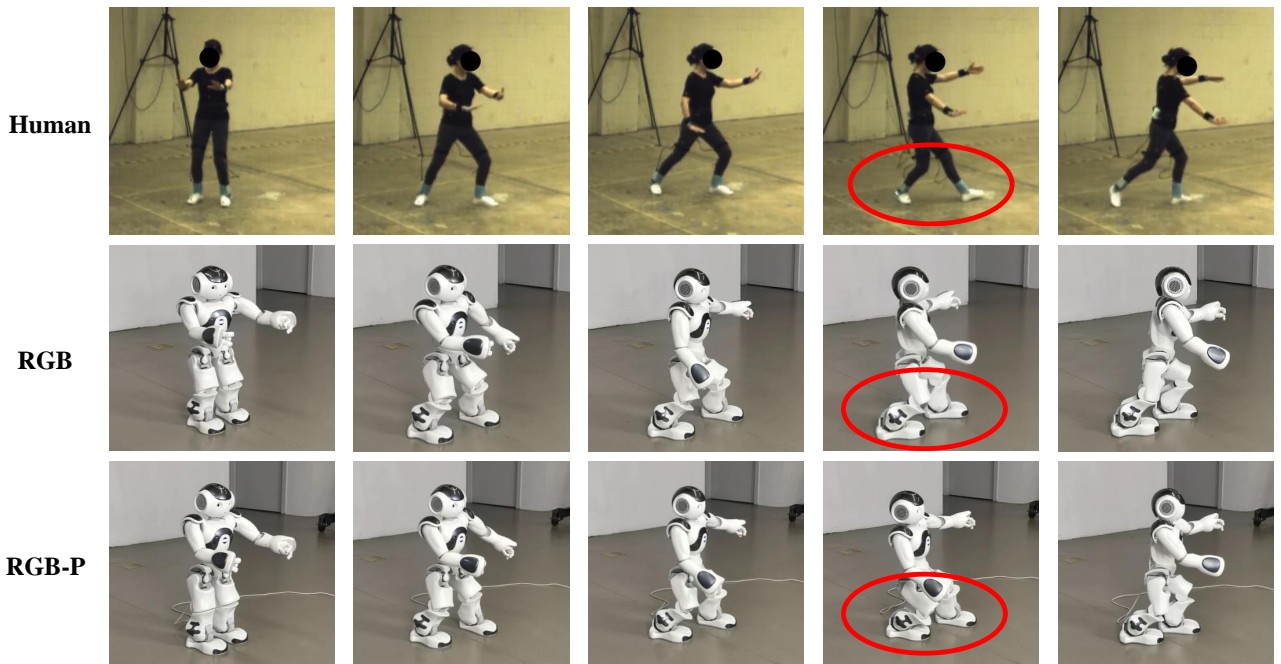

**Figure 6: Comparison of RGB method and RGB-P method given the same pose.**

It can be observed that our method does not improve the accuracy of pose estimation or the execution accuracy of the robot before and after adding pressure. This is understandable because the accuracy of human pose estimation depends on the precision of the RGB-based pose estimation algorithm we employ, and our pose refinement only corrects the CoPs of the human feet, which does not contribute to improving accuracy. It is worth noting that, the performance degradation of RGB-P method is less than 1 mm, which is negligible.

The metric $\mathfrak{E}_{Frechet}$ indicates the similarity between the robot pose $\tilde{\mathbb{P}}_{Robot}$ and the human ground-truth $\tilde{\mathbb{P}}_{Human}$. From the experimental data, it can be observed that the similarity remains almost unchanged before and after the addition of pressure. This suggests that although our refinement leads to a decrease in the accuracy of human pose estimation, this decrease does not significantly affect the similarity of robot execution actions. This also implies that in the task of robot motion imitation, higher accuracy in pose estimation does not necessarily lead to better performance. The motion remapping module plays a more significant role in performance improvement.

## 5.3 Stability Evaluation

As shown in Table 2, we evaluate the stability of our system. The metrics $\mathfrak{E}_{com}$ and $\mathfrak{E}_{cop}$ suggest that integrating pressure data can notably enhance stability. This indicates the effectiveness of our pressure-based pose refinement and support mode discriminator. We also test the stability using human ground-truth (V-MoCap) as input, and the results show that our RGB-P method performed the best. This suggests that the accuracy of human pose estimation does not necessarily lead to improved stability. Instead, it is more crucial to find a mapping relationship that can reasonably establish human and humanoid pose and balance.

## 5.4 Subjective Evaluation

To intuitively demonstrate the effectiveness of human-to-humanoid motion imitation using our proposed method, we conduct tests both in simulation and real-world scenarios.

In Figure 5, we present a motion sequence in PSU-TMM100 alongside corresponding pressure distribution. The imitation results obtained from the pose captured by V-MoCap, RGB, and RGB-P are compared. It is evident that pressure data provides accurate foot-ground contact information, leading to more precise mode recognition. Interestingly, our method even surpasses the V-MoCap approach in terms of action completion, showcasing the advantages of our RGB-P integration. Upon observing instances of falls in both V-MoCap and RGB methods, we note that both of them are caused by misjudgments of the support mode.

The motion imitation performed by a real physical humanoid robot (i.e., NAO) is depicted in Figure 6. For illustrative purposes, only five key frames are selected as examples. It can be observed that our human-to-humanoid motion imitation system can conducted on real robots and achieve good balance between motion similarity and stability. In detail, the methods using RGB-P and RGB are similar in terms of upper-body similarity. But regarding the whole body, the RGB-P method sometimes adjusts leg posture to achieve a support mode more similar to that of humans, resulting in a better imitation effect. As highlighted by the red circles, the human primarily supports her weight on the right foot, whereas the RGB-based method results in a humanoid pose distributing weight across both feet. Conversely, our method achieves a support pattern consistent to that of the human.

## 6 CONCLUSION AND DISCUSSION

In this study, we establish a multi-modal motion mapping system to explore the importance of pressure in humanoid robot imitation. By utilizing RGB and pressure data for humanoid robot motion imitation, we introduce a low-cost and non-intrusive method that enhances stability and balance. Leveraging precise pressure data, we refine the posture of both humans and robots, thereby enhancing their physical consistency. Through experiments in both simulated and real environments, our method demonstrates a significant improvement in stability while maintaining the imitation similarity. However, we must acknowledge that humanoid robot motion imitation still has a long way to go. Making humanoid robots perform routine actions like humans remains highly challenging. We suppose there are several key aspects to consider:

**Robot motion dataset.** Current robot motion imitation faces a challenge due to the limited availability of comprehensive datasets. Most existing datasets are based on human motions, which often include actions beyond robots' capabilities. We believe that in the future, generating synthetic data using computer graphics and physics simulations could broaden the dataset's scope. Transfer learning from existing human motion datasets could also aid in adapting motions to new robot tasks, accelerating learning and improving performance.

**High dynamic motion imitation.** In high-speed dynamic motion imitation, robots face increased complexity in dynamics and kinematics, requiring precise modeling and prediction. This involves understanding interactions between different body parts and the impact of the external environment. Hence, advanced perception and cognition systems are vital. We suppose that integrating various sensors like vision, sound, and touch for comprehensive environmental data is crucial. Additionally, efficient control algorithms leveraging deep learning and reinforcement learning are essential for real-time monitoring and swift adjustments to maintain stability and balance.

**Whole-body motion imitation.** In humanoid whole-body motion imitation, a frequently discussed issue is how to enhance motion stability while ensuring similarity in lower-body actions. For bipedal structures, future focus should also be on maintaining accurate global location control while achieving precise leg imitation. These challenges underscore the need for more advanced sensor and feedback systems, as well as dynamic control algorithms.

**Real-time motion imitation.** Robot teleoperation imposes higher demands on the real-time performance of motion capture and motion control algorithms. We believe that it is highly necessary to develop a low-latency real-time imitation system in the future, which should effectively integrate multiple processes including perception, learning, decision-making, action, and feedback to enhance the capabilities of robot teleoperation.

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
