# OpenReview forum: "Leveraging RGB-Pressure for Whole-body Human-to-Humanoid Motion Imitation"
_acmmm.org/ACMMM/2024/Conference — MM2024 Poster_

### Official Review · Reviewer_Wg96 · 2024-05-24

**Rating:** 3
**Confidence:** 2

**Summary:**

The paper introduces a novel method for enhancing whole-body motion imitation between humans and humanoid robots by integrating RGB cameras and pressure sensors. The paper addresses the limitations of RGB-based methods in accurately capturing human movements due to depth ambiguity, leading to unstable imitation. By incorporating pressure sensors to estimate contact relationships and interaction forces, the proposed RGB-Pressure (RGB-P) system achieves accurate and stable mapping from human body models to robot control parameters.

**Strengths:**

- The paper presents a novel approach of integrating RGB cameras and pressure sensors to enhance whole-body motion imitation between humans and humanoid robots, addressing the limitations of traditional RGB-based methods.
- The proposed RGB-Pressure (RGB-P) system demonstrates improved completeness and stability of robot motion imitation tasks while ensuring motion similarity, as evaluated in both simulation and real-world environments.
- The clarity of the methodology, experimental setup, and results in the paper contributes to its strength, providing a clear understanding of the proposed approach and its implications 3.

**Limitations:**

- The idea of adding a pressure sensor is essentially simple and incremental.
- Let humans wear a pressure sensor may be burdensome. Also, it cannot utilize a large amount of the existing videos on the web.
- Lack of comparison between Physics-aware vision-based motion imitation/learning. For example:
[1] Learning Physically Simulated Tennis Skills from Broadcast Videos, SIGGRAPH 2023
[2] Motion Inbetweening for Physically Simulated Characters, SIGGRAPH Asia 2022
- The paper lacks a comprehensive discussion on the limitations and challenges of the proposed method, particularly in addressing high-speed dynamic motion imitation and whole-body motion stability while ensuring similarity in lower-body actions.
- There is a need for a more detailed comparison with existing methods and approaches in the field of humanoid robot motion imitation, especially in terms of technical correctness and evaluation metrics.

**Suitability:**

2

---

### Official Review · Reviewer_y5Fq · 2024-06-01

**Rating:** 2
**Confidence:** 2

**Summary:**

The paper presents a novel approach for whole-body human-to-humanoid motion imitation, leveraging both RGB camera data and pressure sensor information. The method aims to enhance the stability and accuracy of humanoid robots in mimicking human motions by integrating pressure data to refine pose estimation and ensure a more realistic interaction with the environment. The approach is tested in both simulated and real-world scenarios, demonstrating improvements in motion execution performance.

**Strengths:**

Innovative Multi-modal Approach: The integration of RGB and pressure sensors represents a significant advancement in humanoid robotics, providing a more robust solution for motion imitation by addressing depth ambiguity and enhancing pose accuracy.

Enhanced Stability and Accuracy: The use of pressure data to refine pose estimations and support modes significantly improves the stability and fidelity of the humanoid's movements, making this method particularly valuable for applications requiring high precision.

Comprehensive Evaluation: The paper provides a thorough evaluation of the system, including comparisons with RGB-only methods and testing in both simulated and real-world environments, which illustrates the practical benefits and robustness of the approach.

Potential for Broader Applications: The improved stability and accuracy in humanoid motion could greatly benefit fields such as assistive robotics, entertainment, and complex interaction environments where reliable and safe human-robot interaction is crucial.

**Limitations:**

Complexity and Cost Implications: Integrating pressure sensors, while innovative, introduces additional complexity and potentially higher costs in terms of hardware and maintenance, which might limit widespread adoption.

Limited Novelty in Concept: The concept of enhancing humanoid motion imitation by integrating sensory data is explored in previous research, such as the work presented on Ipman.is.tue.mpg.de, which also discusses the use of intuitive physics for motion understanding. Although the specific application of pressure data alongside RGB data is a valuable contribution, the foundational idea has precedents in the field.

Scalability Issues: Customizing this method to different humanoid platforms and varying environmental conditions could be challenging, potentially affecting the scalability and broad applicability of the proposed approach.

Potential Overfitting to Specific Scenarios: While the system is evaluated comprehensively within controlled environments and specific motion sets, extending its efficacy to a broader range of human activities and less controlled environments may require additional development and validation. This could limit the generalizability of the proposed method.

**Suitability:**

2

---

### Official Review · Reviewer_ooFP · 2024-06-01

**Rating:** 4
**Confidence:** 4

**Summary:**

The paper proposes a novel RGB-Pressure method for enhancing whole-body human-to-humanoid motion imitation, addressing the limitations of depth ambiguity in RGB-only approaches. It integrates RGB cameras for posture capture with pressure insoles to measure underfoot pressure during demonstrations, improving motion stability. A constraint relationship between pressure and pose is analyzed to refine estimations based on support modes and balance, increasing consistency between human and robot motions. The paper outlines a system comprised of pose estimation, motion retargeting, and whole-body control modules. Experiments validate that combining RGB and pressure data enhances robot motion stability and similarity to real human actions, advancing humanoid imitation capabilities.

**Strengths:**

Integrated Sensing Approach: It innovatively combines RGB cameras and pressure sensors (insoles) to overcome depth ambiguity in human motion capture, enhancing the accuracy and stability of humanoid robot imitations.

Consistency Enhancement: The proposed constraint analysis between pressure and pose refines estimated poses for better support and balance, significantly improving the consistency between human and robot motion sequences.

Systematic Framework: Establishes a comprehensive system with clear modules (pose estimation, motion retargeting, and whole-body control) that not only captures real-world poses but also adapts them to humanoid robots' operational parameters.

Performance Validation: Empirical results demonstrate the method's effectiveness in enhancing robot motion execution stability and maintaining imitation similarity to human actions, validated on benchmarks like HumanML3D.

Efficiency & Non-Intrusivity: Offers a low-cost, non-intrusive alternative to expensive or cumbersome motion capture methods, making it practical for broader adoption and enhancing human-robot interaction research.

Application Breadth: Its applicability spans multiple domains like animation, robotics, virtual reality, and rehabilitation, where natural human-like motion imitation is vital for engagement or functional tasks.

**Limitations:**

Depth Data Limitations: The paper should clarify the scope of the dataset used for training and validation, including diversity in activities, age groups, and body types. Limited or biased datasets might affect the generalizability of the model.

Quantitative Metrics for Pressure Integration: While qualitative improvements are mentioned, the paper could benefit from more detailed quantitative metrics evaluating the direct impact of pressure data on motion quality, such as balance, stability improvements in specific poses, or fall-risk scenarios.

Long-term Coherence and Fluidity: The paper focuses on momentary stability but lacks a detailed analysis of long-term motion coherence. How well does the system maintain fluidity and natural transitions over extended sequences, considering the influence of pressure input?

Pressure Sensor Dependence and Calibration: The reliability and calibration of pressure sensors could influence results. The paper should discuss sensor accuracy, drift over time, sensitivity, and how calibration procedures, and potential impact on the system's robustness.

User Study Details: The user study is mentioned but lacks depth. Sample size, demographics, user feedback variability, and a more subjective ratings on perceived naturalness or uncanniness would strengthen conclusions.

Comparison with State-of-the-art: The paper should thoroughly compare against the latest methods integrating additional sensory modalities like force plates or IMU data, not just RGB. Are there any that combine vision-pressure approaches to contrast?

Technical Depth of Analysis: The paper could provide a deeper technical dive into the algorithmic choices, explaining how pressure constraints are modeled, the interaction network architectures refined, and how these decisions were motivated.

Ethical and Safety Considerations: With robots imitating human motion, safety implications arise. Does the paper discuss potential hazards, risk mitigation strategies, or ethical concerns in creating realistic humanoid behaviors?

**Suitability:**

3

---

### Official Review · Reviewer_RygR · 2024-06-01

**Rating:** 4
**Confidence:** 3

**Summary:**

This paper tackles the problem of imitation of human body motions under a monocular camera constraint. The hypothesis is that using a pressure sensor on the foot helps to resolve certain corner cases with monocular-only motion imitation. This is relevant for maintaining stable humanoid control / balancing motions as pressure readings can update the estimates for center of mass and meet constraints from physics. Results from both simuation and real world experiments are shown where adding the pressure modality to the estimation framework seems to improve baseline of RGB-only retargeting.

**Strengths:**

- Leveraging pressure for stablizing and correcting retargeted motions is definitely an interesting idea! Especially considering balancing humanoids and their associated control issues is still being figured out in the community.

- The experiments shown consider a broad range of pose estimation metrics and show improvement along all of them. Using the support mode discrimantor seems to be a simple but effective way to infer leg mappings and make the overall method more robust.

- The paper seems easy to follow and read. Supplementary materials and video also show the key results without much difficulty to the reader.

**Limitations:**

- In terms of experiment metrics, it would be worthwhile to consider for how many trials the control fails (referred to as "unstable" pose in figures/video). Over a sequence of motions to perform, I suppose if RGB-P helps correct the balance, this metric could showcase the success rate across the 100 different sequences in the dataset.

- Adding retargeting of same human motions to different humanoid models (in simulation?) could be also be shown. The current results do show usefulness of RGB-P over RGB but don't seem to motivate why pressure might be a good modality to pursue across generalized humanoid motion retargeting. Another possible drawback is existing large scale human datasets don't seem to include this info.

- Additionally, a fair comparison to other imitation schemes (e.g. RGB-Depth) with the exisiting dataset could be added if possible. Pressure sensors are advocated as "almost non-intrusive", but a depth camera with modern RGB-D based pose estimation methods could also show similar results. It might be beyond the scope in terms of more experiments, so some smaller scale experiment/justification as to why a rgb-depth approach might not work could also help the paper.

**Suitability:**

3

---

### Official Review · Reviewer_G5Xa · 2024-06-01

**Rating:** 4
**Confidence:** 3

**Summary:**

The paper is tackling the whole-body motion imitation challenge. The authors propose system that combining a RGB images with underfoot pressure to improve the posture balance when retargeting the motion to a humanoid robot. The system uses the insole pressure to determine the support mode such as single or dual sides. In the experiment, a PSU-TMM dataset is applied to simulate and compare approaches using different sensing modality quantitatively. In addition to the simulation, they further retarget to a physical robot to validate robot’s balance.

**Strengths:**

The paper is well structured and easy to follow. The flow chat is clear and informative. Their method details the calculation makes the work seem to be reproducible. In the experiment, the pressure does not improve the overall similarity, but their insight on the potential trade-off between balance and similarity due to the target subject’s capacity is rather interesting.

**Limitations:**

Not sure the second contribution that claims “human-to-humanoid motion imitation system with RGB and pressure” is adequate. This work seems does not include details on data collection approach such as camera setup, pressure sensor, but merely use a dataset available to implement the retargeting system.

Minor:
- What’s the initial input S for the first frame in Algorithm 1? Or how to initialize?
- No full name for acronym: IMU

**Suitability:**

3

---

### Meta-Review · Area_Chair_di46 · 2024-07-06

**Recommendation:** Accept (Poster)
**Confidence:** 5

**Metareview:**

This paper received five borderline acceptances from the reviewers, indicating a high level of agreement among them. The area chair supports the acceptance of this paper, as it introduces fresh ideas and goes beyond the common focus on combining vision and language. Instead, it explores the integration of vision with other modalities, specifically pressure sensing in this case. The paper is well-written, and its relevance to multimodal research is strong.

However, as noted by the reviewers, the authors should address the limitations of their work. Specifically, they should discuss the constraints of the scenarios considered, the data used, and potential overfitting issues. Additionally, they should consider comparing their approach with physics-aware methods.